# Lipid Headgroup Charge and Acyl Chain Composition Modulate Closure of Bacterial β-Barrel Channels

**DOI:** 10.3390/ijms20030674

**Published:** 2019-02-05

**Authors:** D. Aurora Perini, Antonio Alcaraz, María Queralt-Martín

**Affiliations:** 1Laboratory of Molecular Biophysics, Department of Physics, Universitat Jaume I, 12071 Castellón, Spain; perini@uji.es (D.A.P.); alcaraza@uji.es (A.A.); 2Section on Molecular Transport, Eunice Kennedy Shriver National Institute of Child Health and Human Development, National Institutes of Health, Bethesda, MD 20892, USA

**Keywords:** bacterial porins, voltage gating, beta-barrel channel, phospholipids, lipid headgroup charge, hydrophobic acyl chains

## Abstract

The outer membrane of Gram-negative bacteria contains β-barrel proteins that form high-conducting ion channels providing a path for hydrophilic molecules, including antibiotics. Traditionally, these proteins have been considered to exist only in an open state so that regulation of outer membrane permeability was accomplished via protein expression. However, electrophysiological recordings show that β-barrel channels respond to transmembrane voltages by characteristically switching from a high-conducting, open state, to a so-called ‘closed’ state, with reduced permeability and possibly exclusion of large metabolites. Here, we use the bacterial porin OmpF from *E. coli* as a model system to gain insight on the control of outer membrane permeability by bacterial porins through the modulation of their open state. Using planar bilayer electrophysiology, we perform an extensive study of the role of membrane lipids in the OmpF channel closure by voltage. We pay attention not only to the effects of charges in the hydrophilic lipid heads but also to the contribution of the hydrophobic tails in the lipid-protein interactions. Our results show that gating kinetics is governed by lipid characteristics so that each stage of a sequential closure is different from the previous one, probably because of intra- or intermonomeric rearrangements.

## 1. Introduction

Membrane channels are the cellular gatekeepers controlling neurotransmission, electrical signaling and the influx and efflux of nutrients and waste products [1,2]. The fine-tuning of these sophisticated transport mechanisms requires that channels behave like active gates. Channel gating refers to conformational transitions between fully conductive (open) states and other (closed) states where transport is partially or totally inhibited. Current fluctuations could occur spontaneously or triggered by a specific stimulus like the binding of ligands, the solution pH or the transmembrane electric potential [2,3].

Historically, the concept of voltage gated-channels has been associated almost exclusively to proteins displaying ion specificity, like sodium (Na^+^), potassium (K^+^), calcium (Ca^2+^), and chloride (Cl^–^) channels [2,4]. These proteins have a highly charged voltage sensor so that a very steep response is found even when small transmembrane potentials appear. Typical switching times between open and close states are quite fast (around milliseconds in Na channels [5]). In contrast, a more gradual response can be found in other protein channels that may remain fully open for long times until a threshold (relatively high) value of the electric potential is met. Also, subsequent closed states could last for seconds or minutes [5]. Examples of this latter class are β-barrel channels in the outer membranes of gram-negative bacteria, mitochondria, and chloroplasts [6]. They typically form large pores with mild ion selectivity that allows for the controlled diffusion of ions and water-soluble metabolites, including respiratory substrates (in mitochondria) and antibiotics (in Gram-negative bacteria). Some pore-forming toxins are also predicted to form β-barrel channels, such as α-hemolysin from *Staphylococcus aureus* or the anthrax protective antigen [7,8].

Voltage gating of β-barrel channels is notoriously complex [9] and the mechanism that rules it is particularly elusive. Gating models point into different directions, from reversible protein denaturation [5], to electrostatic interaction networks [10], or to the concerted movement of a soft or flexible part of the barrel [11,12]. The physiological role of voltage gating in general diffusion porins has been a matter of intense debate. The traditional image of porins depicts them mostly in an open state so that the membrane permeability must be controlled by cellular expression [13]. The possibility that electrical voltage, among other external factors, may regulate the transport across the membrane by promoting the transitions to pore closed states opens a new scenario. In fact, changes in the permeability of outer membrane proteins are found in some antibiotic-resistant pathogenic bacteria [10].

Measurements of membrane potential of *E. Coli* cells at different growth stages display values from −220 mV to −150 mV [14], but these numbers obtained around neutral pH could be reduced by low pH or even flipped to positive polarities (+100 mV) in presence of glutamates and arginines [15]. Of note, measured potentials encompass both the outer and the inner membranes, with no experimental evidence of the individual contribution of each layer. Indirect estimations suggest that the outer membrane accounts for a minor fraction of the total potential drop (<30%) [16], being such values somewhat smaller than the applied voltage required to obtain significant channel gating in vitro [5]. Nevertheless, it is also true that the majority of experiments reporting voltage-induced gating make use of ion channel reconstitution in planar membranes, a protocol specially designed to make the membrane impermeable prior to protein addition [17]. This means that applied voltage values used in this technique cannot be directly translated to the in vivo situation where the membrane is less dense and more permeable.

We consider here the bacterial porin OmpF (Outer membrane protein F) from *E. coli* as a model system for β-barrel channel gating. OmpF is a trimer with three identical subunits (Figure 1), each forming a β-barrel with 16 antiparallel β-strands connected by short (periplasmic side) and long (extracellular side) loops [18]. Each monomer features an hourglass shape due to a central constriction formed by loop L3, which is folded into the barrel. Several studies have shown that OmpF voltage-induced gating changes with the properties of the bathing electrolyte, namely concentration, salt type, and solution pH [5,19,20,21,22,23]. These results suggest that the gating mechanism involves a global reorganization of the channel conformation, rather than just a residue-specific or loop-dependent effect [5,24]. In support of this hypothesis, previous works have demonstrated that extracellular loops as well as loop L3 are not required for channel gating [20,25]. Also, single-residue mutations have shown different mild effects on the channel gating, but no one proved to be essential [26,27,28,29].

Within the concept of a general contribution of the channel structure into the gating process, a change of the mechanical or electrostatic properties of the lipid membrane should affect the channel gating. Indeed, a recent study has shown that lipids modulate OmpF gating [13]. We delve into the lipid role by exploring separately how the hydrophilic polar heads and their hydrophobic tails participate in the lipid-protein interactions. We demonstrate that headgroup charge and acyl chain composition have strong effects on OmpF gating, revealing a complex interaction of the protein with the surrounding lipid.

## 2. Results

OmpF forms wide channels that allow the transport of small ions, metabolites, and antibiotics. In the open state, the channel has a high conductance (~4 nS in 1 M KCl) and a slight preference for cations over anions [30,31]. When inserted in planar lipid membranes, OmpF displays a complex voltage-dependence, most times with a stepwise closure that reveals the trimeric character of the channel but also occasionally with fast flickering showing spontaneous closures and reopenings of any of the three monomers [5].

Here we focus exclusively on the channel sequential closings. To this end, after the insertion of a single OmpF trimer, we apply a high voltage (±200 mV) and evaluate the time needed for the open channel to become partially or totally closed. Figure 2 shows representative current traces of typical experiments, for single OmpF channels inserted in either a neutral diphytanoyl-phosphatidylcholine (DPhPC) lipid membrane or negatively charged diphytanoyl-phosphatidylserine (DPhPS) one. Diphytanoyl is a saturated acyl chain [32] commonly used in planar membrane electrophysiology. For both positive (Figure 2a) and negative (Figure 2b) applied potentials, each stepwise transition yields a reduction of one third of the open channel current (indicated as 0), which reflects the (almost) complete blockage of one of the monomers in the trimer (indicated as 1, 2, 3 for one, two, or the three monomers closed, respectively). The OmpF channel closure is reversible, so the monomers reopen after the high voltage is turned to zero.

From the visual inspection of current traces, it becomes apparent that a negatively charged membrane enhances OmpF gating at both voltage polarities. Due to the stochastic nature of the channel gating [33,34], the time spent in each state is a random variable so that strong statistics are necessary to obtain reliable results. Thus, for each lipid composition and applied voltage, we collected more than 250 events. See Materials and Methods for more details on the event analysis. Figure 3 shows logarithmically binned histograms [35] of the time needed to close the first monomer (τ_0_) under a positive (a) or negative (b) applied voltage. Histograms can be fitted by a single exponential (solid lines in Figure 3a,b) indicating that, at least for the closure of the first monomer, OmpF gating can be approximated by a two-state Markov process [33,34].

Figure 3c shows the characteristic times obtained from the fittings of Figure 3a,b. When inserted in DPhPC membranes, it takes around 1 s for the channels to close its first monomer after a potential is applied, in line with previous studies [5]. For negatively charged headgroups—DPhPS—τ_0_ is reduced to less than 0.5 s. Of note, the dependence on voltage polarity is opposite in each lipid. τ_0_ is larger at negative potentials in DPhPC and at positive ones in DPhPS.

Next, we checked how acyl chain composition affects OmpF voltage-induced gating. To this end, we evaluated gating kinetics of OmpF in membranes formed by a mixture of DPhPC and dioleoyl-phosphatidylcholine (DOPC) (1/1). DOPC, a lipid with markedly different properties compared to DPhPC, was chosen to maximize any possible effect of the lipid hydrophobic core on OmpF gating [32,36]. We used a lipid mixture to enhance single channel insertion, which was found to be significantly less efficient in pure DOPC. Results are shown in Figure 4, comparing pure DPhPC with the mixture DPhPC/DOPC (1/1) (labelled as DOPC). We found that τ_0_ distributions for DPhPC/DOPC can be fitted again by a single exponential (solid lines in Figure 4a,b), meaning that DPhPC and DOPC do not form separate domains but a somewhat homogeneous mixture.

The characteristic times obtained from the fittings of Figure 4a,b are shown in Figure 4c. τ_0_ values in DOPC mixtures are lower than in pure DPhPC. Note that we are using a mixture of 50% DOPC, so a milder effect compared to pure lipids is expected. The effect of acyl chains in OmpF gating is clearly dependent on voltage polarity: at positive potentials, there is no significant effect of 50% DOPC, while at negative applied voltages there is a twofold reduction of τ_0_. As in the case of lipid charge, a change in hydrophobic composition reverses the dependence on voltage polarity.

Up to the present point we have limited our kinetics analysis to the closure of the first monomer. Each one of three OmpF monomers is thought to be functionally identical and independent in the open state and when interacting with small molecules like antibiotics [30,37]. To analyze how monomers behave during the gating process, we evaluated not only the time necessary to close the first monomer (τ_0_), but also the second (τ_1_) and third ones (τ_2_). In our approach τ_1_ and τ_2_ begin when the previous closure occurs (not when voltage was applied), so that the time to close the full trimer is τ_0_ + τ_1_ + τ_2_. As shown in Figure 5, gating kinetics for the second and third closures are not well described by single exponential distributions, but the best fittings for τ_1_ and τ_2_ correspond to two-exponential distributions. Within Markov kinetics [2], a response with 2 characteristic times points to the existence of 3 different states. All characteristic times obtained are shown in Table 1, including neutral DPhPC, negatively charged DPhPS, and the heterogeneous acyl chain DPhPC/DOPC, for positive and negative applied voltages. In DPhPC, dwell times for slow kinetics (higher values of τ_1_ and τ_2_), are reasonably similar to τ_0_ within the experimental error, at both polarities. Lower values of τ_1_ and τ_2_ may correspond to fast kinetics because they are much lower (0.01–0.3 fold) than τ_0_ and reasonably close to flickering events (~ms). The results with a negatively charged membrane (especially at positive voltage) and with DPhPC/DOPC mixture (at both voltage polarities) are significantly different. First, dwell times are not comparable anymore because the differences between τ_0_, τ_1_ and τ_2_ exceed by far the experimental error. Secondly, the different kinetics in τ_1_ and τ_2_ do not show a definite pattern in relation to τ_0_ (0.01–20-fold variations can be found). Altogether, this results clearly show a complex scenario for the second and third monomer closures.

## 3. Discussion

The OmpF channel is the major porin in *E. coli* mediating the transport of ions and water-soluble metabolites across the outer membrane, including nutrients and antibiotics. We have shown exhaustively (Figure 2) that the voltage-induced gating dramatically reduces the effective ion transport across the channel and eventually halts the translocation of larger molecules. The implications for membrane permeability control are captivating. General diffusion porins could behave like active valves instead of as passive holes and therefore, membrane permeability could be controlled in more subtle ways than just protein expression.

We pay special attention to how lipid–protein interactions regulate channel currents. The interplay between lipid environment and any given protein is complex because it may imply reorganization of both protein and membrane conformations [38]. *E. coli*’s outer membrane has a complex lipid composition, with phospholipids at one leaflet and lipopolysaccharides at the other [39]. However, for the sake of clarity, we have explored separately how hydrophilic lipid heads and the hydrophobic tails participate in gating kinetics using membranes with symmetric lipid compositions.

We first focused on the effect of lipid charges. We demonstrate that a change in the membrane from neutral diphytanoyl-PC to negatively charged diphytanoyl-PS significantly promotes OmpF channel closure. The open-channel lifetime is reduced more than two fold in diphytanoyl-PS, both at positive and negative applied potentials (Figure 3). To elucidate how lipid charge participates in channel gating, it is convenient to stress that electrostatic interactions are important modulators of voltage-induced closures of OmpF. Either pH changes or mutations in the protein charged residues at the extracellular side yield radical changes in gating kinetics [5,13]. Atomic Force Microscopy images suggest that both low pH and applied voltage induce the same conformational change of porin OmpF, namely a rotation of the extracellular domain at the rim of the β-barrel [19]. Molecular Dynamics simulations show that the extracellular surface loops interact strongly with lipid headgroups, especially with charged ones [13], what would explain our findings.

Next, we evidence that changes in the hydrophobic composition of the membrane also lead to different gating kinetics (Figure 4). We compare diphytanoyl-PC with dioleoyl-PC, thus keeping the same zwitterionic lipid head and changing the two saturated 16-carbon acyl chains into two longer ones (18 carbons) with one unsaturation each [32,36]. The effect of dioleoyl-PC is to reduce the duration of the open state, a similar effect of adding a negatively charged lipid head (compare Figure 3c with Figure 4c). Several reasons could be behind such findings. A change in the hydrophobic phase of the membrane alters the mismatch between the protein and the membrane hydrophobic core [40] but also the protein stability and plasticity [41]. Also, changes in the elastic stress of lipid packing could appear because dioleoyl-PC has lower interfacial area per lipid as well as lower lipid volume, compared to diphytanoyl-PC [32,36].

The complexity of lipid-protein interactions in which multiple factors play a role becomes evident by confronting the present study to previous using the same protein. Recently, Liko et al. reported that membrane lipid composition modulates OmpF channel closure in experiments performed at pH 4 [13]. They compared diphytanoyl-phosphatidylcholine (DPhPC) with palmitoyl-oleoyl-phosphatidylglycerol (POPG), thus varying lipid headgroup charge and acyl chain at the same time. Longer gating times obtained in POPG were ascribed only to the interactions between lipid heads and protein residues [13]. These findings seem to be at odds with our experiments with PS shown in Figure 2 and Figure 3. Note however, that both sets of experiments (Ref [13] versus the present one) differ in at least two crucial features: solution acidity (pH 4 versus pH 6), and hydrophobic composition (palmitoyl-oleoyl versus diphytanoyl). In addition, it is necessary to consider that the particular chemistry linking channel residues to lipid headgroups may change even between polar heads having the same charge (PG versus PS). Indeed, recent investigations show that gating in VDAC, another β-barrel channel, can be ruled by chemically-specific interactions between protein residues and lipid heads via intermolecular hydrogen bonds [42].

Finally, we have investigated the gating of the three sequential closures of OmpF. The fact that all three monomers display similar current levels for the closed states (Figure 2) does not necessarily imply that they have also analogous kinetics. If the overall gating kinetics were ruled by superposition of independent two-state Markov processes (open - close), the time necessary to close two identical monomers would be, on average, 2x the time it takes to close one monomer; likewise, the time necessary to close three identical monomers will be 3x the time needed to close one monomer. Hence, we would expect τ_0_ ~ τ_1_ ~ τ_2_. However, this is not the case of OmpF channel. As shown in Figure 5, neither do we find a unique relaxation for each closed state nor the characteristic times τ_0_, τ_1_, τ_2_ are identical in any case.

We hypothesize that multiple relaxations found within 1 and 2 levels and the dissimilarity between relaxation times in different levels have to do with monomer distinctiveness and independence. Thus, the fact that the closure of the first monomer can be fitted by single exponentials suggests an identical gating mechanism for all the three open monomers in level 0. This would agree with precedent studies stressing that all monomers perform equally in the fully open state [30,37]. In contrast, τ_1_ and τ_2_ are better represented by two-exponential distributions meaning that monomers in level 1 and 2 are not identical as regards gating. In other words, to get multiple closures it matters which subunit(s) closed previously, since it determines whether a fast or a slow kinetics will appear for the next closure. Moreover, from the fact that τ_0_, τ_1_ and τ_2_ are different it follows that, in each stage of a sequential process, the gating kinetics is different from the previous closure. This suggests that during the gating process there are either intra-monomeric reorganizations or intermonomeric interactions in which the lipids participate altering the overall kinetics.

To recapitulate, our results demonstrate the importance of lipid membrane composition in the mechanism of voltage-induced gating in large bacterial β-barrel channels. This adds to the number of central channel properties such as open channel current, selectivity, and interaction with partners that depend crucially on membrane composition [12,43,44,45,46,47,48,49]. In particular, we show unambiguously that lipid headgroup charge is an important modulator of OmpF gating, but it is not the only factor. On the contrary, the membrane hydrophobic core also plays an important role in promoting OmpF closure. These results support a model of β-barrel gating in which the whole protein is involved in a general conformational change in the presence of a transmembrane voltage. Such reorganization probably leads to changing interaction between monomers in which lipid characteristics are crucial. Understanding the mechanism of channel gating and identifying the controlling factors is essential given the potential regulatory role of voltage-gating in bacterial homeostasis and antibiotic uptake.

## 4. Materials and Methods

Planar membranes were formed from apposition of two monolayers [17] across orifices with diameters of ~100 μm on a 15-μm-thick Teflon partition using different mixtures of diphytanoyl-phosphatidylcholine (DPhPC), diphytanoyl-phosphatidylserine (DPhPS) and dioleoyl-phosphatidylcholine (DOPC), from Avanti Polar Lipids (Alabaster, AL). Lipids were dissolved in pentane at 5 mg/mL and amounts of 10–20 μL were added to each chamber. The orifices were pre-treated with a 1% solution of hexadecane in pentane. Membranes were formed at least 10 min after hexadecane and lipid addition to ensure pentane evaporation. The OmpF protein was a generous gift of Sergey M. Bezrukov, NIH, Bethesda (MD), USA. To achieve a single-channel insertion, 0.1–0.5 μL of protein at 1 ng/μL in 0.1 M KCl and 1% (v/v) OctylPOE (Alexis, Switzerland) were added to the 2 mL aqueous phase at the *cis* side of the membrane chamber after membrane formation. Channel insertion was enhanced by applying an external voltage of ±50–100 mV, but this has no effect on the channel orientation once inserted. In fact, control experiments show that the conductance of OmpF for negative voltages is almost always (>95%) higher than for positive ones indicating that the channel inserts predominantly in a particular orientation [50]. The electric potential was applied using Ag/AgCl electrodes in 2 M KCl, 1.5% agarose bridges assembled within standard 250 μL pipette tips. The potential was defined as positive when it was higher at the side of the protein addition (the *cis* side of the membrane chamber), whereas the *trans* side was set to ground. An Axopatch 200B amplifier (Molecular Devices, Sunnyvale, CA, USA) in the voltage-clamp mode was used to measure the current and applied potential. The signal was digitized at 50 kHz sampling frequency after 10 kHz 8-pole in-line Bessel filtering. The chamber and the head stage were isolated from external noise sources with a double metal screen (Amuneal Manufacturing Corp., Philadelphia, PA, USA). All experiments were performed with 1 M KCl solutions buffered with 5 mM HEPES at pH 6. Channel gating was analyzed by applying a high voltage (±200 mV) to trigger gating events in relative short times (of the order of seconds), so enough events could be recorded to gather sufficient statistics. Channel dwell times were collected using the *Event Detection* tool from Clampfit 10.6 (Molecular Devices, Sunnyvale, CA, USA). All the analyses shown in the manuscript focus only on sequential 0-1-2-3 closures discarding traces that showed spontaneous reopenings. Once totally closed, the voltage was put to 0 mV for ~10 s before a new experiment with the fully open channel began. Also, events shorter than 20 ms were ignored to avoid the eventual fast flickering, which was not the focus of this study. For each dwell time, events were gathered from different channel insertions in at least three independent experiments. Characteristic times were calculated by fitting logarithmic exponentials to logarithmically binned histograms (10 bins per decade) [35]. Exponential fittings used the Simplex search method and the four available minimization methods, calculating a mean ± SD from their average. Significance between experimental conditions was checked with SigmaPlot 12.3 (Systat Software, Inc.) by a one-way ANOVA test. In case of significant differences, it was used a Holm–Sidak *post hoc* test for pair-wise comparison. Data were checked for normality and homoscedasticity with Shapiro–Wilk and Levene tests, respectively. See Supplementary Material for more details.

## Figures and Tables

**Figure 1 ijms-20-00674-f001:**
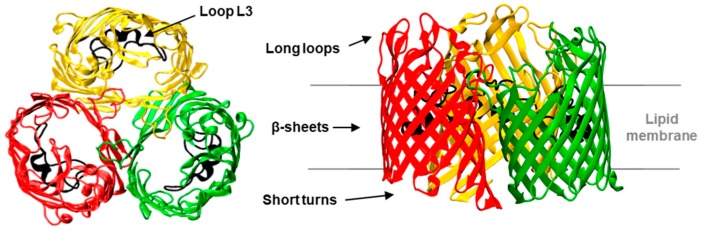
Three-dimensional structure of the OmpF (Outer membrane protein F) channel. View from the extracellular side (left panel) and side view (right panel) of the OmpF homotrimer (PDB code 2OMF). Each monomer is folded as a 16-stranded antiparallel β-barrel connected by eight long loops facing the extracellular side and eight short turns facing the intermembrane space. Loop 3 (shown in black) folds inside the barrel, creating a constriction zone (~1 nm) at half the length of the monomer.

**Figure 2 ijms-20-00674-f002:**
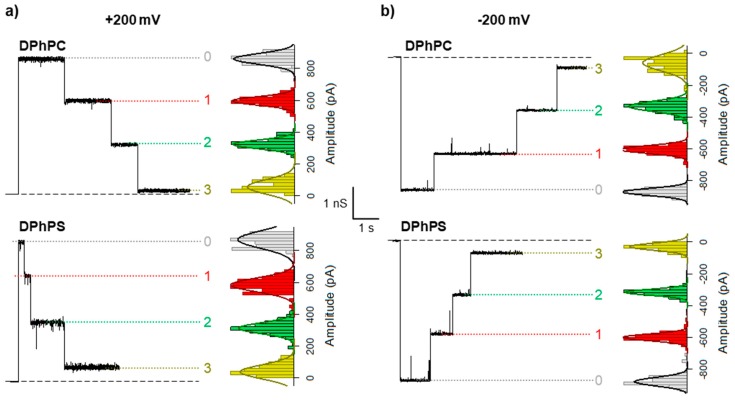
OmpF sequential closing depends on lipid headgroup charge. Representative current traces of a single OmpF trimer inserted in a neutral diphytanoyl-phosphatidylcholine (DPhPC, upper panels) or a negatively charged diphytanoyl-phosphatidylserine (DPhPS, lower panels) membrane, with the typical step-wise transitions after a high positive (**a**) or negative (**b**) voltage is applied. Next to each trace, the amplitude distributions are shown for each state (bar graph) with a Gaussian fitting (solid lines).

**Figure 3 ijms-20-00674-f003:**
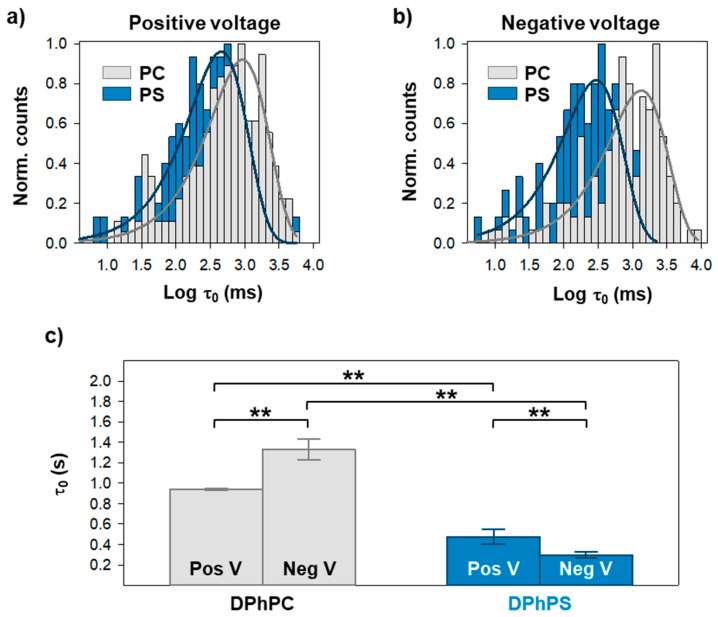
A negatively charged membrane accelerates closure of first OmpF monomer. Logarithmically binned histograms of the time the channel spends in the open conformation under a positive (**a**) or a negative (**b**) applied voltage when inserted in a neutral or negatively charge membrane, as indicated. Solid lines are exponential fittings. The characteristic times obtained from exponential fittings are shown in (**c**). See Appendix A for the distributions in (**a**) and (**b**) displayed individually. In (**c**), significance between mean closure times was determined with a one-way ANOVA significance test. A Holm–Sidak *post hoc* test was used for pair-wise comparison (**: *p* < 0.01). See Materials and Methods for more details.

**Figure 4 ijms-20-00674-f004:**
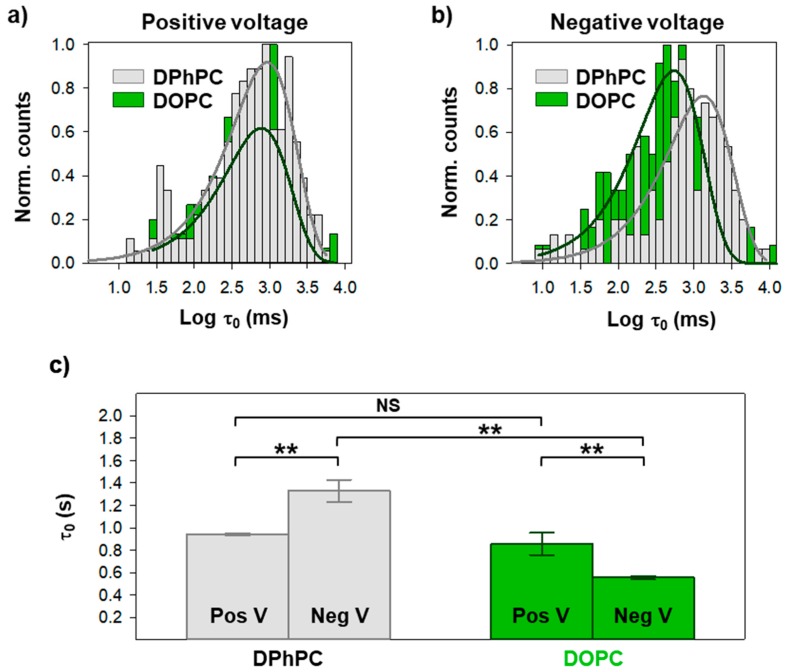
Change in acyl chain composition modulates closure of first OmpF monomer. Logarithmically binned histograms of the time the channel spends in the open conformation under a positive (**a**) or a negative (**b**) applied voltage when inserted in membranes composed of pure DPhPC or DPhPC/DOPC (diphytanoyl-phosphatidylcholine/dioleoyl-phosphatidylcholine 1/1, labelled as DOPC), as indicated. Solid lines are exponential fittings. The characteristic times obtained from exponential fittings are shown in (**c**). See Appendix A for the distributions in (**a**) and (**b**) displayed individually. In (**c**), significance between mean closure times was determined with a one-way ANOVA significance test. A Holm-Sidak *post hoc* test was used for pair-wise comparison (NS (not significant): *p* > 0.2; **: *p* < 0.01). See Materials and Methods for more details.

**Figure 5 ijms-20-00674-f005:**
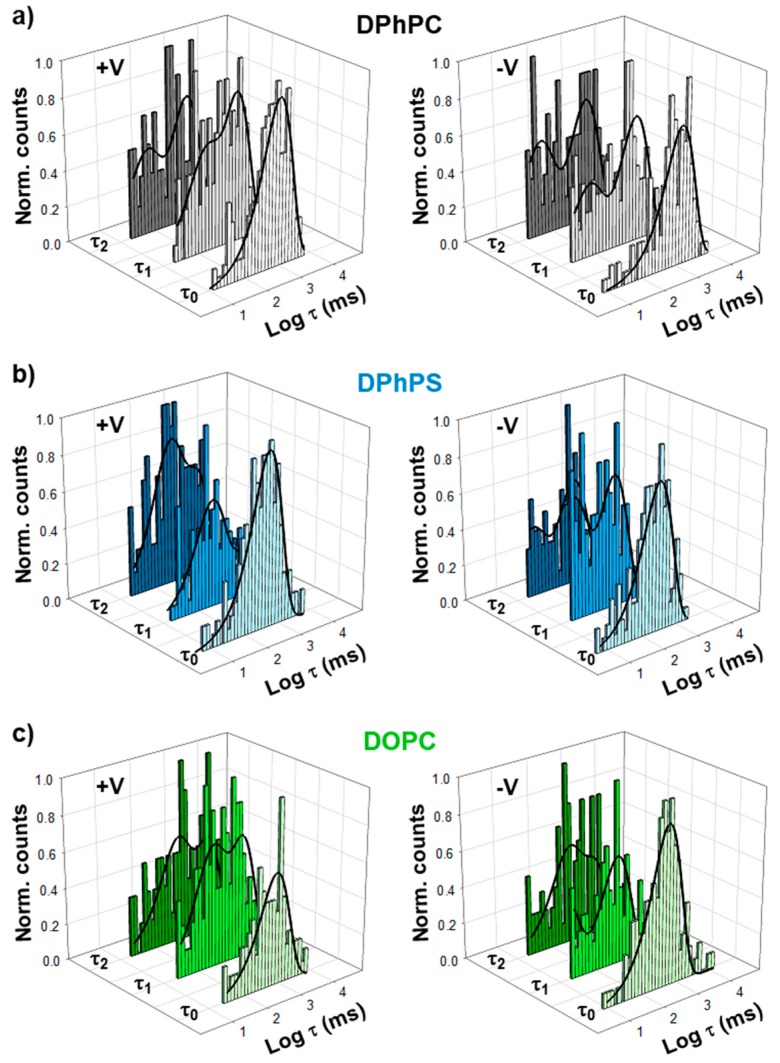
Gating kinetics for the second and third monomer closures display higher complexity than the first one. Logarithmically binned histograms of the time the channel spends before closing the first (τ_0_), second (τ_1_), or third (τ_2_) monomers under a positive (left panels) or a negative (right panels) applied voltage when inserted in membranes composed of DPhPC (**a**), DPhPS (**b**), or DPhPC/DOPC (1/1) (**c**). Solid lines are exponential fittings with one (τ_0_) or two (τ_1_, τ_2_) terms. See Appendix A for all distributions displayed individually.

**Table 1 ijms-20-00674-t001:** Characteristic times obtained from the fitting of histograms in Figure 5 for the time the channel spends before closing the first (τ_0_), second (τ_1_), and third (τ_2_) monomers. τ_0_ was fit with a single exponential, while τ_1_ and τ_2_ were best described by a two-exponential distribution. For τ_1_ and τ_2_, the relative weight of the exponentials is indicated in parenthesis, calculated as the relative area defined by each exponential.

**Positive V**	**τ_0_ (ms)**	**τ_1_ (1) (ms)**	**τ_1_ (2) (ms)**	**τ_2_ (1) (ms)**	**τ_2_ (2) (ms)**
DPhPC	942.0 ± 10.9	34.2 ± 51.7 (0.31)	944.3 ± 295.5 (0.69)	42.1 ± 11.4 (0.34)	868.4 ± 156.2 (0.66)
DPhPS	476.3 ± 73.5	189.4 ± 45.2 (0.67)	1246.1 ± 322.9 (0.33)	222.6 ± 9.6 (0.49)	1662.8 ± 93.3 (0.51)
DOPC	856.0 ± 98.0	56.4 ± 95.0 (0.41)	1099.4 ± 529.9 (0.59)	360.6 ± 91.6(0.44)	3394.4 ± 502.5 (0.56)
**Negative V**	**τ_0_ (ms)**	**τ_1_ (1) (ms)**	**τ_1_ (2) (ms)**	**τ_2_ (1) (ms)**	**τ_2_ (2) (ms)**
DPhPC	1329.8 ± 100.2	14.3 ± 26.8 (0.33)	933.4 ± 397.1 (0.67)	11.0 ± 19.8 (0.40)	889.5 ± 342.6 (0.60)
DPhPS	296.9 ± 31.0	6.0 ± 8.6 (0.43)	314.5 ± 114.9 (0.57)	336.1 ± 102.6 (0.35)	245.9 ± 316.6 (0.65)
DOPC	554.0 ± 15.2	69.0 ± 116.2 (0.41)	439.4 ± 25.6 (0.59)	391.1 ± 188.2 (0.46)	6169.0 ± 7397.1 (0.54)

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
