# Peer review of "Lipid Headgroup Charge and Acyl Chain Composition Modulate Closure of Bacterial β-Barrel Channels"

_ijms, 2019, doi:10.3390/ijms20030674_

Round 1
Reviewer 1 Report
This study examines the effect of the charge of lipid head groups and chain length of the reconstitution membrane on kinetics of voltage dependent OmpF channel closures. DPhPC (neutral diphytanoyl-PC) versus DPhPS (charged diphytanolyl-PS) and DPhPSC (16 carbon) with DOPC (18 carbon). They found the voltage polarity, charge on the lipids and acyl length all changed the kinetics of closure. This indicates a role for lipids in regulation of the channel. The article is well written but the following needs to be addressed:
When describing the use of the lipids for the first time (for example on line 105 for DPhPC), please give the name of the lipid and indicate the chain length. Otherwise it is only given on line 207. Alternatively, the names could be given in the abbreviations section. Please do the same with the other lipids. The descriptions in line 207-219 could then be shortened.
On line 256 the authors refer to one of the remaining channel subunits as "who" in italics. Rather than "who", "which of the remaining subunits" would probably be better, since "who" refers to a person.
Author Response
Answer to Reviewer 1 is provided in the attached Word file.

Reviewer 2 Report
This study investigates the modulation of OmpF channel function by the membrane composition. This channel was reconstituted in planar lipid bilayers, and when a strong negative or positive voltage was applied, the channel closed within a few seconds. In the present study it is shown that changing the head group of the membrane lipids from one having no net charge (PC) to a negatively charged group (PS) accelerated the first closing step of the trimeric channel assembly. Modification of the lipid chain of the membrane lipids also affected these kinetics. The trimer closes in three steps. The kinetics of the second and third closing are complex and appear also to be affected by the membrane composition.
Major points
1. The experiment with the replacement of the head groups of PC to PS gives an information on the dependence of the kinetics on the head group charge. The experiments in which the lipid part is changed are however much more difficult to interpret. Phytanic acid is not a major physiological component of membranes. It differs from oleate (to which it is replaced in the experiments), not only in the length, but also in chemical properties. In the study, DPhPC is partially replaced by DOPC. In order to be able to interpret such data, it would have made more sense to change only one parameter, e.g. chain length or saturation or chemistry, at a time.
2. Presentation and analysis of the data. The authors state that for the analysis of the kinetics of the first closing event, >250 events were analyzed per condition. It is however never shown from how many independent experiments these data come, and it is not clear to me how the error (shown in Fig. 3c, 4c and table 1) is calculated, and whether this is an experimental variation, or the error of the fit. The number of independent experiments, and of events should be indicated for all conditions. The distributions in Fig. 5 are not easy to see, and it seems that for some conditions, the number of events may be quite low, and might need additional experiments. The authors draw conclusions about differences (that the kinetics are faster with negative head charge, etc.). They should use and provide statistics to support the significance of the differences that are at the basis of these statements. In table 5, the second and third closing are described by 2 exponentials each. The relative weight of the exponentials needs to be indicated. Figs. 3 and 4 a-b show overlapping distributions in which the test condition (PS in Fig. 3, DOPC in Fig. 4) is shown behind the control distribution. With this way of presentation, only part of the distribution of the test condition is visible. Could the distributions be shown in a way that shows them in full?
3. Analysis of second and third monomer closures: This analysis is presented, but no clear conclusions are made. Some of the taus seem to be ill defined, with error values greater than the tau value itself. Elaborate better the conclusions that you can draw from these analyses.
Specific points
1. Introduction, lines 41-44, “These proteins…” This sentence is not clear and should be reformulated. “Typical closed states last around milliseconds”. This depends largely on the voltage. Voltage-gated Na channels for example are closed for long times at resting membrane potentials.
2. Order of closings. In Fig. 2, ideal sequential closings are shown. In reality it is likely that sometimes or quite often a channel may go back and forth between two levels (for example between level 1 and 2) before going to a lower level. Were for the analysis of Fig. and Tab. 5 only experiments with sequential 0-1-2-3 level closings used, or were events also included when they resulted from more complicated patterns? This should be indicated. From reading the manuscript, it seems that the times measured for tau1 and tau2 are the time from the first to the second closing (tau1) and from the second to the third closing (tau2). In the text they are however described as the kinetics of the second and third closing. Therefore I would expect from this description that they represent rather the time from the beginning of the voltage step to the second and third closing. This should be clarified in the manuscript. It is also said in the text that channels could be reopened by putting the voltage back to 0 mV. Indicate, for which duration the voltage was put to 0 mV before a new negative or positive voltage step was applied.
3. Figs. 3c and 4c: Is it more important for the authors to show an effect of the sign of the voltage, or to show an effect of the membrane composition? The bars should be grouped accordingly. In the text it appeared that the membrane composition was more important. In this case it would be more appropriate to place in Fig. 3c the “Pos V” conditions with “DPhPC” and “DPhPS” side by side and do the same for “Neg V”, and do similarly in Fig 4c.
4. Several time it is discussed that the control of the OmpF channel opening by voltage may be clinically relevant. To support this point, it is necessary to indicate the normal voltage difference across the outer membrane of bacteria, and discuss possible mechanisms that may change it.
5. Line 287, is “250 ml pipette tips” a typing error?
Author Response
Answer to Reviewer 2 is provided in the attached Word file.

Reviewer 3 Report
Everyone can read my comments.
Overall Recommendation: Accept after minor revisions.
Review Report:
Brief summary
Perini et al examine the role of lipid head-group charge and of lipid acyl-group composition in the voltage dependent closing of the OmpF protein, a resident of the outer membrane of E.coli. The other feature of OmpF that figures into their results is that the protein is a trimer of three identical subunits. In their study, they find that both lipid head-group charge and fatty acid composition alter the voltage induced closing times of the OmpF protein. With the lipids tested, they find that the time of the first of closing stimulated with a positive 200 mV potential is routinely 942 ms with a single Gaussian fit of all first closing events in the DPfPC lipid bilayer. However, the next two closings share no such single Gaussian closing characterization, but require two Gaussians to fit the observed additional closing times of the second and third monomers in the OmpF protein. These times are quite different from that of the first closing except the second Gaussian for both t1 and t2 is not too different from the first closing time in DPfPC. The other lipid bilayers tested are DPhPS and a heterogeneous mix of DPhPC and DOPC. The first closing times for OmpF in DPhPS, DOPC mix, and DPhPC with the positive voltage stimulation are 476, 856 and 942 ms. The first closing responses with -200mV stimulation are 297, 554 and 1330 ms. Apart from noting the differences of each bilayer’s effect on these first closing times, additional information is gathered about subsequent closing times of the other two monomers.
Broad comments: I find this an interesting adventure into understanding the operation of OmpF, which at one point was regarded as an open pore, and permeation was controlled by OmpF expression. Clearly, the results of Perini et al. as well as the work reported by Liko et al. indicate there is more to the story. The first thought that comes to mind about the work is that there is no reference to the lipid organization of the E.coli outer membrane in which OmpF works. My prejudice is that the membrane is quite asymmetric. Wonpil Im in the Biophysical Journal in 2014 claims that the outer membrane has phospholipids in the inner membrane leaflet and no phospholipids in the outer leaflet, which he claims is composed primarily of lipopolysaccharide. My sense is that this oligosaccharide component would be a very interesting component to test for the closing times of OmpF. Now all that is required is the production of this asymmetric membrane, which is the next study for Perini et al. It is a bit of a problem to have studied the closing times in symmetric membranes, but the role of the head group charge and the length of the fatty acid is a good start. My guess is Perini et al should go to shorter fatty acid chains rather than longer ones to encourage the incorporation of OmpF in the synthetic bilayers. I say this with no experience, so I admire their work in using this technology to produce a membrane with OmpF as a component for study. The ability to form asymmetric membrane bilayers using their technology might not be possible, but perhaps it could be tried. When I look at Montal’s model of the machine for doing it, it seems possible. Having the two monolayers in two troughs on opposite sides of the Teflon partition and then forcing them up to cover the opening? How practical that is, I am not sure, but it sure would be fun to see the results of their experiments if the OmpF would incorporate into the asymmetric bilayer. Does the voltage-forced incorporation of OmpF into the bilayer cause right side up or upside down OmpF’s or both? Also not addressed: What is +200 mV application to the outside (cis) of the membrane if OmpF goes in with the bottom first? Then a -200 mV application on the outside as well? Not sure but it seems the top side should be ground and the bottom side (trans) should be –X to correspond to what is going on with the E. coli? I think a little discussion of what is known about the reconstitution into the bilayer would go a long way to sort all this out. I think a bit of discussion of the strategy then in choosing the ground and the applied voltage would be appropriate in regard to the orientation of the OmpF in the bilayer even if to justify the -/+ 200 mV. Why such a large voltage? If only to observe an effect, then it should be so stated. Inside living eukaryotic cells hyperpolarization rarely goes to
-200mV. What is known about the voltage differences across the outer membrane? Is
there any?
In any case, I recommend the Im paper for consideration of the authors:
Biophys J. 2014 Jun 3; 106(11): 2493–2502.
doi: 10.1016/j.bpj.2014.04.024
PMCID: PMC4052237
PMID: 24896129
E. coli Outer Membrane and Interactions with OmpLA
Emilia L. Wu,† Patrick J. Fleming,‡ Min Sun Yeom,§ Göran Widmalm,¶ Jeffery B. Klauda,|| Karen G. Fleming,‡∗∗and Wonpil Im†∗
Specific comments: The OmpF figure on page 2 rather than referring to the left panel as “front”, should it be “from the inside looking out”? This is a strange way of looking at a protein embedded in a membrane encasing a bacterium. I would favor the “outside looking in”, which would require flipping the right monomer over from right to left. Then it would present in the same orientation as the side view next to it. Line 89: “We explore separately how the lipid hydrophilic polar heads and their hydrophobic tails…” We “deep”? Line 92: for “differentiated” use “modulating”? Figure 2. I would lobby to label the tops +200 mV and -200 mV rather than positive and negative voltage. Figure 3: be sure to label the x axis of the right panel. I am not sure all readers will know what DPh represents. It is an unusual lipid fatty acid, so I am not sure how best to enlighten the reader. Why was DPhPS chosen other than it seems to take in OmpF in reconstitution applications? Line 208: I would use “reduced more than two fold”. Line 267: “hydrophobic core also plays”… Line 290: “The signal was digitized…”.
Author Response
Answer to Reviewer 3 is provided in the attached Word file.

Round 2
Reviewer 2 Report
The authors have responded to most of the points that I raised. Two points still need to be addressed:
On line 64, "Measurements of membrane potential...". The cited reference discusses the potential difference between the inside of E. coli and the outside. This is not the potential difference across the outer membrane. The potential difference across the outer membrane would be important for OmpF gating. The authors should indicate here any indication of such a potential difference, and if such evidence does not exist, they should state it.
Statistics (Figs. 3-4): Using only a one-way ANOVA is not appropriate, since in each of these figures, 4 conditions are compared. More appropriate statistics need to be used.
Author Response
The answer to Reviewer 2 has been uploaded as a PDF.
